# Rapid Identification of 3,6′-Disinapoyl Sucrose Metabolites in Alzheimer’s Disease Model Mice Using UHPLC–Orbitrap Mass Spectrometry

**DOI:** 10.3390/molecules27010114

**Published:** 2021-12-25

**Authors:** Jiaqi Yuan, Han Wang, Yunting Wang, Zijian Wang, Qing Huo, Xueling Dai, Jiayu Zhang, Yaxuan Sun

**Affiliations:** 1Beijing Key Laboratory of Bioactive Substances and Functional Foods, Beijing Union University, Beijing 100191, China; 18235577790@163.com (J.Y.); wanghan42075@163.com (H.W.); wyunting2021@163.com (Y.W.); huo_q2002@aliyun.com (Q.H.); xueling@buu.edu.cn (X.D.); 2Beijing Research Institution of Chinese Medicine, Beijing University of Chinese Medicine, Beijing 100029, China; helloffiresilver@gmail.com; 3School of Pharmacy, Binzhou Medical University, Yantai 264003, China; zhangjiayu0615@163.com

**Keywords:** Alzheimer’s disease, 3,6′-disinapoyl sucrose, UHPLC–Orbitrap mass spectrometry, metabolites, metabolic pathway

## Abstract

Alzheimer’s disease (AD) is a degenerative disease of the central nervous system characterized by the progressive impairment of neural activity. Studies have shown that 3,6′-disinapoyl sucrose (DISS) can alleviate the pathological symptoms of AD through the activation of the cAMP/CREB/BDNF signaling pathway. However, the exact biochemical mechanisms of action of DISS are not clear. This study explores metabolism of DISS in an AD mouse model, induced by the microinjection of a lentiviral expression plasmid of the APPswe_695_ gene into CA1 of the hippocampus. After gavage administration of DISS (200 mg/kg), the kidneys, livers, brains, plasma, urine, and feces were collected for UHPLC–Orbitrap mass spectrometry analysis. Twenty metabolites, including the prototype drug of DISS, were positively or tentatively identified based on accurate mass measurements, characteristic fragmentation behaviors, and retention times. Thus, the metabolic pathways of DISS in AD mice were preliminarily elucidated through the identification of metabolites, such as ester bond cleavage, demethoxylation, demethylation, and sinapic acid-related products. Furthermore, differences in the in vivo distribution of several metabolites were observed between the model and sham control groups. These findings can provide a valuable reference for the pharmacological mechanisms and biosafety of DISS.

## 1. Introduction

Alzheimer’s disease (AD) is a progressive neurodegenerative disorder that occurs mostly in the elderly and is characterized by insidious onset and slow but irreversible progression. The pathology of AD is characterized by insoluble plaques resulting from an accumulation of extracellular amyloid β (Aβ) and intracellular neurofibrillary tangles. These pathologies can contribute to impaired memory and cognition, resulting in severe mental decline and behavioral abnormalities that negatively impact quality of life [1,2]. In particular, Aβ is the protein hydrolysate of human amyloid precursor protein (APP) following sequential cleavage events by β- and γ-secretases [3,4].

Numerous studies have shown that APP was associated with AD. Conformational changes and transport caused by abnormal APP modifications can alter secretion of lytic enzymes, resulting in dysregulated APP processing and excessive Aβ production, which in turn can contribute to neuronal dysfunction [5]. In aging astrocytes and hippocampal neurons, the accumulation of mutant APP and Aβ induced mitochondrial malfunction, and abnormal neuronal autophagy, resulting in neuronal death and cognitive decline [6,7]. The APP gene is located on human chromosome 21 and its mRNA can be processed by selective splicing into three major isoforms: APP_695_, APP_751_, and APP_770_. Among these isoforms, APP_751_ and APP_770_ are primarily expressed in glial cells, and APP_695_ is most abundantly expressed during neuronal differentiation [8], the overexpression of APP_695_, the gold standard for generation of classical AD models, and is widely used in research focused on AD prevention and treatment [9,10,11,12]. In this study, an AD model was generated by injecting an APPswe_695_ gene lentiviral expression plasmid into the CA1 region of mice hippocampi.

Kaixin San (KXS), a well-known traditional Chinese medicine prescription, has been used for thousands of years in China to treat dementia and amnesia. Studies have shown that KXS effectively cured AD and depression in a rat model. Specifically, KXS treatment resulted in an increased expression of neurotransmitter-regulating enzymes and synaptic proteins in the nervous system, which promoted synapse formation, enhanced neurite outgrowth in response to the increased expression of the nerve growth factor, and enhanced neprilysin expression, which promoted degradation of Aβ_42_. These biochemical changes resulted in neuroprotection and increased cognitive ability in rats [13,14,15,16]. Polygalae Radix (*Polygala tenuifolia* Wild.) is a component of KXS and is mainly used to relieve neurasthenia, forgetfulness, dementia, and depression. Studies have shown that 3,6′-disinapoyl sucrose (DISS, Figure 1), an oligosaccharide ester isolated from Polygalae Radix, can act as an antidepressant, and exert neuroprotective and antioxidant effects, via activation of the cAMP/CREB/BDNF signaling pathway [17,18,19,20,21]. However, the effects and potential mechanisms of action of DISS on AD have not been characterized. Furthermore, the biotransformation of DISS in vivo has not been elucidated.

The aim of this study is to investigate differences in DISS metabolite profiles between normal mice and AD mice overexpressing APPswe_695_. Differences in metabolic profiles were used to characterize in vivo DISS metabolic pathways, which is a critical step in the determination of potential neuroprotective mechanisms of DISS against AD. Ultra-high performance liquid chromatography coupled with Orbitrap mass spectrometry (UHPLC–Orbitrap–MS/MS) was used to detect and identify DISS metabolites in plasma, urine, feces, and vital organs of mice in each group. Due to its superior sensitivity and high accuracy, this technology is extensively used not only in pharmaceutical analysis, food chemistry, and environmental monitoring, but also for chemical composition analysis, compound quantitation, and pharmacokinetic studies [22,23,24,25].

## 2. Materials and Methods

### 2.1. Chemicals and Reagents

Chengdu Gelipu Biotechnology Co., Ltd. (Chengdu, Sichuan, China), supplied DISS with purity greater than 98% as determined using UHPLC–Orbitrap analysis. A lentiviral plasmid expressing the APPswe_695_ gene was provided by Professor Huang Hanchang of the Beijing Key Laboratory of Bioactive Substances and Functional Food. Nissl staining solution was purchased from Beyotime Biotechnology Co., Ltd. (Shanghai, China). In addition, HPLC grade methanol, formic acid, and acetonitrile were purchased from Beijing RUIZHX Tech Co., Ltd. (Beijing, China). ProElut^TM^ C18 solid-phase extraction (SPE) cartridges (500 mg/3 mL, 50 µm, 60 Å), for the pretreatment of mouse biological samples, were purchased from Dikma Technologies Co., Ltd. (Foothills Ranch, CA, USA). All other chemicals were of analytical grade and were purchased from the Beijing Chemical Works (Beijing, China).

### 2.2. Animals

Thirty-two male ICR mice (19–21 g) were purchased from Beijing Weitong Lihua Experimental Animals Company (Beijing, China). The animals were housed in a controlled environment with the temperature held at 21–25 °C, relative humidity held at 55–70%, and a 12 h light/dark cycle. After one week of acclimatization, all mice were divided randomly into the following 3 groups: control group (for blank biological samples, *n* = 10), sham control group (*n* = 10), and model group (*n* = 12). All animal facilities and experimental procedures were approved by the institutional Animal Care and Use Committee of the Center of Functional Inspection of Health Food, College of Applied Science and Humanities, Beijing Union University. The animal experiments approved by this committee were executed from December 2019 to February 2020 and numbered as 2019-12. There were no significant differences in body weight or food intake among the experimental groups.

### 2.3. Microinjection and Surgery

An AD mouse model was established using an intra-hippocampal injection method. Mice were anesthetized by intraperitoneal injection of pentobarbital sodium (80 mg/kg) and placed on a stereotaxic apparatus (RWD Life Science Co., Ltd., Shenzhen, China). The bregma of each mouse was exposed by cutting a small incision and adjusted to the same horizontal plane, and bilateral burr holes were drilled according to Paxinos’ and Franklin’s The Mouse Brain Atlas [26]. The stereotaxic coordinates were as follows: anteroposterior, −0.20 cm from bregma; medial/ventral, ±0.22 cm; and dorsal/ventral, −0.25 cm. All mice in the model group were injected bilaterally with 0.6 μL of lentiviral plasmid expressing the APPswe_695_ gene. The mice in the sham control group were injected with 0.6 μL of blank plasmid. Each mouse was injected through a single burr hole for 10 min and the syringe was kept in CA1 for 5 min to ensure sufficient infusion of plasmid. Then, the incision was sutured and wiped with iodophor. Following the procedure, penicillin (40,000 U) was injected intramuscularly for 3 days.

### 2.4. Novel Object Recognition Test (NORT)

The NORT assesses learning and memory capability using the natural curiosity of mice to explore new things. This experiment consists of the following three stages, as shown in Figure 2: (1) Adaptation period: mice were separately placed in a dark empty device and allowed to walk freely for 10 min (bottom is 50 cm × 50 cm, and the height is 60 cm). After adaptation for each mouse, the device was wiped with 75% alcohol to eliminate odor. (2) Familiarity period: two identical cylindrical objects (object A1 and A2, bottom is 6 cm, height is 12 cm) were fixed on specific locations in the device to prevent movement, and the mice were placed and allowed to move freely for 5 min. To eliminate of the olfactory cues, the device and objects were thoroughly cleaned with 75% alcohol after each trial. (3) Test period: after 24 h, 1 cylindrical object (A1) was replaced with a cuboid object (object B, bottom is 6 cm × 6 cm and the height is 7 cm), and mice were placed in the apparatus and allowed to explore for 5 min. The behavior of the mice during the tests was recorded with a camera. The time (T) during which the mice explored the two objects within 5 min, was recorded manually using 2 stop watches, and the recognition index (RI) was calculated using the following expression: T_B_/T_A2_ + T_B_. Exploration was defined as a distance less than 1.0 cm between the mouse nose and the object when the mouse was facing the object.

### 2.5. Morris Water Maze (MWM) Test

The MWM test used, assesses the spatial learning and memory capability of rats and mice. The device consists of a water pool (180 cm in diameter and 60 cm in height) and a movable black platform with a diameter of 4 cm. The water temperature in the device was maintained at 21 ± 2 °C, and the water surface was 1.5 cm higher than the platform. To make the water opaque and to provide a good contrast with the white color of the ICR mice, black non-toxic ink was added to the water. A camera was installed above the tank, and connected to the SMART digital tracking system (Version 3.0, Panlab, Cornella, Spain). Distal visual cues were fixed to the corners of the room. The pool was divided into four quadrants and the platform was placed in the center of one quadrant. The MWM test was executed according to the method of Steadman et al. and Focke et al. [27,28]. Mice were trained three times per day. Each mouse was placed daily in different quadrants, facing the wall of the pool. In each trial, the mouse was given a maximum of 60 s to escape to the platform and remain on the platform for 15 s. Mice that failed to find the platform within 60 s were guided to the platform, where they remained for 15 s. Training was performed for four days, and the platform was never moved. The latency to escape was the time for the mouse to find the platform. Memory retention was assessed on day 5, using a spatial probe test with the platform removed. Each mouse was tested once, and was placed facing the pool and allowed to swim freely for 60 s. The swimming tracks of the mice were recorded using the SMART digital tracking system. The times and distances of mice in the target quadrant (recorded as percentages of total time), and the first time to reach the platform were recorded to evaluate the spatial learning ability of mice.

### 2.6. Sample Collection and Preparation

DISS was suspended in 0.5% sodium carboxymethyl cellulose solution. The sham control group and model group mice were orally administered a dose of 200 mg/kg body weight, and the control group was given an equivalent volume of 0.5% CMC-Na solution. Prior to gavage, all mice were fasted for 12 h and were allowed water ad libitum.

#### 2.6.1. Plasma Sample Collection

After oral administration, all mice were placed in metabolic cages, and blood samples were taken from the suborbital venous plexus at 0.5, 1, 1.5, 2, 4, and 6 h, and blood was collected from the same mouse twice at most. The blood volume of each mouse was not less than 100 μL and not higher than 200 μL; therefore, the total amount at each time was not less than 300 μL and ensured that no deaths occurred during this process. At each time point, 3 mice were selected in sequence for blood collection. When all mice collected blood once, the mice were sequentially selected for blood until the blood samples at six time points were collected. Blood samples were placed in centrifuge tubes containing 10 μL of heparin sodium anticoagulant. The samples were held at room temperature for 15 min, and then centrifuged at 3000 rpm (4 °C). Plasma samples from the same group were pooled, which resulted in blank plasma (control group) and test plasma (sham control group and model group) samples.

#### 2.6.2. Urine Sample Collection

Urine samples were collected from 0–24 h by the urine tubes of metabolic cages following gavage, and samples from each group were pooled. Then, samples were centrifuged at 3500 rpm and 4 °C for 15 min, and the supernatants were blank urine and urine after oral DISS.

#### 2.6.3. Feces Sample Collection

The feces samples of each group were collected from 0–24 h through the feces canister of the metabolic cages, and rinsed with deionized water when a small amount of urine was dropped in. Fecal samples were dried and ground, 1 g of feces of each group was mixed with deionized water (5 mL), and the samples were ultrasonically extracted for 60 min. The fecal extracts were centrifuged at 3500 rpm (4 °C) for 15 min to separate supernatants. Finally, blank feces sample and test feces samples (sham control group and model group) were obtained.

#### 2.6.4. Plasma, Urine, and Feces Samples Preparation

The plasma, urine, and feces samples of each group were extracted using SPE. The SPE cartridges were pre-activated with 5 mL of methanol and 5 mL of deionized water (2.5 mL each time, twice), and all plasma (the blood volume in each group was between 0.8–1.0 mL) was added. Then, the cartridges were eluted with 5 mL (2.5 mL each time, twice) of deionized water and 3 mL of methanol in sequence. The methanol eluate was collected and dried under nitrogen at room temperature. The dried residue was redissolved in 80 µL of 10% acetonitrile and centrifuged for 30 min (14,000 rpm, 4 °C). The supernatant was used for subsequent analysis. The urine (1 mL) and feces (1 mL) samples were prepared using the same process procedure as the plasma samples.

#### 2.6.5. Tissue Sample Collection and Preparation

Following sample collection, the mice from each group were sacrificed. Four mice in the sham control group and model group were perfused with 0.9% physiological saline and 4% paraformaldehyde. Then, the brains were stripped and fixed in paraffin for Nissl staining. Brain, liver, and kidney tissue was collected from the remaining mice and rinsed with precooled physiological saline and dried with filter paper. A total of 0.08–0.14 g of liver was taken from each mouse, and the livers of each group were pooled, and homogenized in saline (1 mL, 4 °C), and extracted with ethyl acetate (1 mL) 3 times using ultra-sonication. The ethyl acetate fractions were collected and centrifuged at 14,000 rpm (4 °C) for 30 min. The supernatant was used for UHPLC–MS/MS analysis. The kidneys and brains of each group were treated in the same process procedure as the liver samples.

### 2.7. Nissl Staining

Coronal sections of brain tissue (thickness 4 µm) were dewaxed in xylene for 5 min, 3 times each. Then, the slices were washed with 100% ethanol for 5 min, 90% ethanol for 2 min, 70% ethanol for 2 min, and distilled water for 2 min, in sequence. The sections were incubated in Nissl stain at 37 °C for 10 min, washed twice with distilled water (2 s each time), and washed with 95% ethanol for 5 s. Then, the sections were dehydrated twice using 95% ethanol (2 min each time) and xylene for 5 min. The sections were made transparent through incubation in fresh xylene for 5 min, and the sections were sealed with neutral resin for visualization. Images were collected using a microscope (Nikon, Tokyo, Japan) and quantified using ImageJ version 1.50 (National Institutes of Health, Bethesda, MD, USA).

### 2.8. Instruments and Analytical Conditions

Rapid separation of DISS metabolites by UHPLC was performed on a Thermo Vanquish Flex Binary RSLC platform (Thermo Fisher Scientific, Waltham, MA, USA). Samples were separated on a Thermo Accucore aQ C18 (150 × 2.1 mm, 2.6 µm; Thermo Fisher Scientific, Waltham, MA, USA) maintained at 40 °C. The mobile phase consisted of 0.1 formic acid water (A) and methanol (B). Gradient elution was performed as follows: 0–3 min, 0–20% B; 3–8 min, 20–35% B; 8–14 min, 35–70% B; 14–18 min, 70–75% B; 18–21 min, 75–95% B; 21–28 min, 95–97% B; 28–30 min, 97–100% B and 30–45 min, 100% B. The flow rate was 0.3 mL/min and the injection volume was 3 µL.

Analysis of the metabolites was performed using a Q Exactive Plus combined quadrupole Orbitrap mass spectrometer (Thermo Fisher Scientific, Waltham, MA, USA). Analysis was performed using electron spray ionization in both positive and negative ion modes, and the specific parameters were set as follows: capillary temperature, 350 °C; spray voltage, 4 kV/3.5 kV (positive/negative); sheath gas, 50 arb; auxiliary gas, 10 arb; and probe heater temperature, 320 °C. Full-scan analysis, from *m*/*z* 150–1500 with a resolution of 70,000, was performed under positive and negative ion modes. Collision-induced dissociation was performed using an isolation width of 2 Da, and 30% maximum of collision energy.

### 2.9. Data Processing and Statistical Analyses

The behavioral test data were analyzed using IBM SPSS Statistics 22.0 (SPSS, Inc., Chicago, IL, USA), and the results were expressed as means ± standard deviation. For the MWM test, repeated measure ANOVA was used to analyze escape latency in the positioning navigation experiment. The data of NORT and spatial probe test in MWM were analyzed by one-way ANOVA, and the Student’s t-tests with the Bonferroni correction were used to determine significant differences between groups. The alpha level was set at *p* < 0.05. Experimental data were plotted using Origin 9.1 software (Originlab Corporation LLC., Northampton, MA, USA).

The collected mass spectrometry data was processed and analyzed using Thermo Xcalibur 3.1 software (Thermo Scientific, Bremen, Germany). Peaks with intensity over 10,000 were selected so that as many fragments as possible could be acquired. The parameters of the formula predictor were set as follows: C (0–50), H (0–100), O (0–50), S (0–5), N (0–5), and ring double bond (RDB) equivalent value (0–15). The maximum mass error was 5 ppm.

## 3. Results

### 3.1. Novel Object Recognition Test (NORT)

Learning, recognition, and short-term memory capability of the experimental mice was determined by the recognition index (RI: the ratio of the time that the mice spontaneously explored novel objects to the total time) [29]. The mean RI of the model group was 0.5131 (standard deviation 0.1046), and that of the sham control group was 0.6545 (standard deviation 0.1148). Through *t*-test analysis, it was found that the RI of the model group was significantly lower than that in the sham control group (*p* = 0.007 < 0.01), indicating that the memory ability and curiosity of the model mice were significantly reduced (*p* < 0.01).

### 3.2. Morris Water Maze Test

During the training period, the escape latency of the two groups gradually decreased, but the escape latency of the model group was always longer than that of the sham control group, and there was a significant difference in the escape latency between the two groups within 4 days of training (*p* < 0.05, Figure 3A). In the spatial probe test, in comparison to the sham control group, the model group had a shorter dwell time in the target area with a shorter swimming distance (*p* < 0.05, Figure 3B,C), and a distinctly longer time to reach the platform for the first time (*p* < 0.05, Figure 3D). These results suggested that the learning and memory abilities of AD model mice were significantly impaired. Comparing the swimming trajectories of the two groups, the mice in the sham control group showed obvious exploratory behavior, and the model group mice had adherent movement during the spatial probe test. The movement trajectories of one mouse in each group were shown in Figure 3E,F. In conclusion, the spatial memory ability of the model group mice were significantly reduced than the sham control mice.

### 3.3. Neuronal Pathology in the Hippocampal Region

The Nissl body is a large granule containing the endoplasmic reticulum and ribosomes, and is an important site for protein synthesis. In damaged neurons, the Nissl body will dissolve or disappear. Nissl staining was performed to confirm the destruction of neuronal structure or apoptotic conditions in the brain tissue of AD model mice. The results showed that the neurons in the CA1 region of the hippocampus, in the sham control group, were tightly arranged and had a normal morphological structure, while the model group had a reduced number of neurons in the same region with sparse distribution (Figure 4). These findings showed that the hippocampal region of the mice in the AD model was severely damaged and neuronal apoptosis had occurred in the brain.

### 3.4. Analysis of Mass Fragmentation Behavior of DISS

The chromatographic and mass spectral information for DISS was collected using UHPLC–Orbitrap mass spectrometry, to further determine its fragmentation pattern (Figure 5), which was important for the identification and characterization of its metabolites. In the positive ion mode, an [M + Na]^+^ ion at *m*/*z* 777.22015 (C_34_H_42_O_19_Na, −1.10 ppm) was present for DISS with a retention time of 10.10 min. In the ESI–MS^2^ spectrum, DISS was associated with the characteristic ions at *m*/*z* 409 and *m*/*z* 391, resulting from glycosidic bond breakage, and subsequent C–O bond cleavage resulted in the appearance of *m*/*z* 207.

In the negative ion mode, DISS had an [M − H]^−^ ion at *m*/*z* 753.22412, and cleavage of the C–O bond and the glycosidic bond resulted in characteristic fragments at *m*/*z* 547 and *m*/*z* 367, respectively. The fragment ions at *m*/*z* 223, were generated by cleavage of the C–O bonds of the *m*/*z* 753, 547 and 367 fragments, and the loss of H_2_O from the *m*/*z* 223 fragment resulted in an abundant *m*/*z* 205. The characteristic ion at *m*/*z* 205, sequentially lost CH_3_ to sequentially form the product fragments at *m*/*z* 190 and *m*/*z* 175, which in turn lost CO and H_2_O to generate the characteristic fragments at *m*/*z* 164 and 149, respectively.

### 3.5. Identification of DISS Metabolites

Thermo Xcalibur 3.1 was used for the analysis of total ion chromatograms (TICs) collected, using UHPLC–Orbitrap MS from mouse plasma, urine, feces, and tissues. Comparison chromatograms from the sham control group with those from the model group, resulted in a total of 20 metabolites detected in the positive and negative ion modes. The metabolic pathway of DISS is shown in Figure 6, and the chromatographic and mass spectrometric information for all metabolites is summarized in Table 1.

The analytic M0, with a retention time of 10.10 min, had an [M + Na]^+^ ion at *m*/*z* 777.22101 (C_34_H_42_O_19_Na, −0.309 ppm) and an [M − H]^−^ at *m*/*z* 754.23148 (C_34_H_41_O_19_, 2.157 ppm). These ions were also present in the DISS reference standard. In its ESI–MS^2^ spectrum, the major fragment ions *m*/*z* 205, 391, 409, and 547 were consistent with the fragmentation pattern of the DISS reference standard. These results allowed for positive identification of M0 as DISS.

Metabolite M1 eluted at 5.78 min, and had an [M + Na]^+^ ion at *m*/*z* 571.16290 (C_23_H_32_O_15_Na, −0.773 ppm), which was 206 Da lower than the sodium adduct of DISS. Fragment ions at *m*/*z* 391 and *m*/*z* 409 were observed in the ESI−MS^2^ spectra of M1. Furthermore, M1 showed fragmentation patterns in both positive and negative ion modes similar to those of DISS. Based on these finding, M1 was presumed to be the product of cleavage of the glycosidic bond of DISS (6′-sinapoyl sucrose).

In the positive ion mode, metabolites M2 and M6 eluted at 6.61 min and 8.46 min, respectively. Each metabolite was associated with protonated molecular ions [M + H]^+^ at *m*/*z* 227.09128 (C_11_H_14_O_5_, error within ±0.60 ppm). In ESI−MS^2^ spectra, an [M + H − H_2_O]^+^ ion at *m*/*z* 209 was unexpectedly detected, and unique fragments at *m*/*z* 181, 177, and 167 demonstrated that M2 and M6 were products of C–O bond cleavage and the reduction of DISS, and were positional isomers.

Metabolites M3 and M18 had the elution times of 7.96 and 16.04 min, respectively, and had [M + H]^+^ ions at *m*/*z* 227.12766 (C_12_H_18_O_4_, ±0.55 ppm). In the MS/MS spectra, characteristic fragment ions were generated at *m*/*z* 209.11690 (C_12_H_17_O_3_), *m*/*z* 191.10643 (C_12_H_15_O_2_), and *m*/*z* 149.09598 (C_10_H_13_O). These results indicated that M3 was a methylation product of M2, and that M18 was a positional isomer of M3.

Metabolite M4 eluted at 8.13 min, and had an [M − H]^−^ ion at *m*/*z* 339.10873 (C_16_H_20_O_8_, 3.792 ppm). The fragment ions at *m*/*z* 259.09747, 229.08693, and 134.03729 were produced by the successive losses of 2H_2_O + CO_2_, CH_2_O and C_6_H_7_O. Metabolite M9 generated [M + H]^+^ ion at *m*/*z* 341.12158 (C_16_H_20_O_8_, error with −4.438 ppm) and eluted at 11.21 min. In the ESI−MS/MS spectra, characteristic ions at *m*/*z* 323.09164, 238.19051, and 205.08562 were observed. These results indicated that M4 and M9 were DISS demethoxylation products.

Metabolite M5, which eluted at 8.14 min, had a sodium adduct [M + Na]^+^ at *m*/*z* 735.17440 (C_31_H_36_O_19_Na, error within 0.1914 ppm), which was 42 Da less than the mass of the sodium adduct of DISS. This result suggested that M5 was a demethylation product of DISS, with loss of C_3_H_6_. The characteristic fragments in the ESI−MS/MS spectra were *m*/*z* 277.11856, 249.12337, and 343.18710, which were generated by C–O bond cleavage. Loss of water from the *m*/*z* 343 ion, resulted in an *m*/*z* 325 ion. In the negative ion mode, the ion at *m*/*z* 711.17834 [M − H]^−^ (C_31_H_36_O_19_, 2.299 ppm) underwent C–O bond fragmentation, resulting in fragments at *m*/*z* 321.04388, 294.13840, and 241.08717 in the ESI−MS/MS spectrum. These results further suggested that M5 was likely a demethylation product of DISS.

Metabolite M19, which had an [M − H]^−^ ion at *m*/*z* 663.17542 (C_27_H_36_O_19_, −1.938 ppm), and had a molecular weight 68 Da less than that of DISS, was likely the product of a loss of C_7_H_6_ from DISS. The ESI−MS^2^ spectra contained *m*/*z* 391.20743, *m*/*z* 323.22034, and *m*/*z* 255.23296 ions. Fragmentation of a C–O bond, likely resulted in the conversion of the *m*/*z* 323.22034 ion to the *m*/*z* 255.23296 ion. Therefore, M19 was determined to be a demethylation product of DISS, and an isomer of M5.

Metabolite M7 had an [M − H]^−^ ion at *m*/*z* 223.06116 [M − H]^−^ (C_11_H_12_O_5_, 1.060 ppm), which eluted at 8.57 min. The ESI−MS/MS spectra contained a characteristic ion at *m*/*z* 207.06502, generated by loss of CO that was tentatively identified as sinapyl. The ion at *m*/*z* 207, sequentially lost CH_3_, CO, and CH_3_, resulting in fragments at *m*/*z* 192.04141, *m*/*z* 164.04666, and *m*/*z* 149.02347, respectively. Metabolite M7 was identified as sinapinic acid. Metabolite M8 had a protonated ion [M + H]^+^ at *m*/*z* 243.08733 (C_11_H_14_O_6_, 1.921 ppm) and a retention time of 10.99 min. The molecular weight of M7 was 18 Da greater than that of M7. Based on these data, M8 was likely a hydroxylation product of sinapic acid.

M15 had a deprotonated ion at *m*/*z* 223.09724 (C_12_H_16_O_4_, 3.382 ppm) and a retention time 14.64 min. The C–O bond was broken and CO_2_ was lost, following collision-induced cleavage, resulting in fragments at *m*/*z* 122.10566 and *m*/*z* 179.12711 in the ESI−MS/MS spectra, respectively. These fragments indicated that M15 was a demethylation product of DISS.

Metabolites M10 and M14, with retention time at 12.75 and 14.48 min, generated [M − H]^−^ ions at *m*/*z* 371.1349 and *m*/*z* 371.13239 (C_17_H_24_O_9_, error with 3.399 ppm and −3.418 ppm), which were 32 Da more than those of M9 and M4. In the negative ESI−MS/MS spectra, the ion at *m*/*z* 291.16016 (C_17_H_23_O_4_) was the cleavage product of the DISS glycosidic bond of M14, and the subsequent loss of CO_2_ resulted in the generation of the *m*/*z* 247.17033 (C_16_H_23_O_2_) fragment. M10 was characterized by cleavage of the glycosidic bond at *m*/*z* 209.11717 (C_12_H_17_O_3_), and, in positive ESI−MS/MS, C–O, and C–H bonds, were broken to produce ions at *m*/*z* 377.27933, *m*/*z* 271.12012, and *m*/*z* 359.26834. Metabolites M10 and M14 were likely products of cleavage of the DISS glycosidic bond, and were positional isomers.

M11 produced a sodium adduct at *m*/*z* 395.13120 [M + Na]^+^ (C_17_H_24_O_9_, −0.388 ppm) and a retention time of 13.39 min. The C–O bond of M11 underwent fragmentation, and the characteristic ions at *m*/*z* 121.10120 and *m*/*z* 217.15601 were generated. According to fragment ions and retention time, M11 was identified as a product of cleavage of the DISS C=O bond.

M12 had an [M + H]^+^ ion at *m*/*z* 241.10716 (C_12_H_16_O_5_, 0.456 ppm), a retention time of 13.90 min, and was 16 Da greater in mass than M7. The characteristic product ions at *m*/*z* 123.116687 and *m*/*z* 147.11676 were detected in the ESI−MS/MS spectra. These results indicated that M12 was the product of the reduction of methylated sinapinic acid.

Metabolite M13 eluted at 13.52 min and had an [M − H]^−^ ion at *m*/*z* 227.09253 (C_11_H_16_O_5_, error with 4.976 ppm), which was 4 Da greater in mass than M7. The product ions at *m*/*z* 209.11829 and *m*/*z* 165.12851 were due to the loss of H_2_O and CO_2_, respectively. In positive ion mode, M13 was detected at 13.56 min, and was associated with an [M + Na]^+^ ion at *m*/*z* 251.08891 (C_11_H_16_O_5_, −0.338 ppm). Fragmentation resulted in cleavage of the C–O bond, which produced product ions at *m*/*z* 167.08630, *m*/*z* 173.13144, and *m*/*z* 141.09071 in the ESI−MS/MS spectra. These data resulted in identification of M13 as a sinapinic acid addition product.

Metabolite M15 had a retention time of 14.64 min, and had an [M − H]^−^ ion at *m*/*z* 223.09724 (C_12_H_16_O_4_, 3.382 ppm), which was 2 Da less than the mass of M12. The main fragment ions in the ESI−MS/MS spectra were *m*/*z* 122.10566 and 179.12711. These results allowed for the tentative identification of M15 as a reduction product of M12.

Metabolite M16, which eluted at 15.32 min, had an [M + H]^+^ ion at *m*/*z* 727.28077 (C_34_H_46_O_17_, error with −1.947 ppm). In the ESI−MS/MS spectra, the characteristic ion at *m*/*z* 427.10205, further fragmented to produce ions at *m*/*z* 409.09137 and 391.08096 through sequential loss of H_2_O. Further fragmentation of *m*/*z* 511.12332, resulted in ions at *m*/*z* 481.11246 and 349.07037 through sequential loss of CH_2_O moieties. In the negative ion mode, M16 eluted at 15.30 min and had an [M − H]^−^ ion at *m*/*z* 725.26600 (C_34_H_46_O_17_, 1.2745 ppm). Fragmentation resulted in the loss of CO moieties, to produce the *m*/*z* 353.06680, *m*/*z* 545.12988, *m*/*z* 413.08752, and *m*/*z* 407.07764. These fragments were consistent with those observed in the positive ion mode. These data indicated that M16 was a reduction product of DISS.

M17 eluted at 15.69 min, and had an [M + H]^+^ ion at *m*/*z* 729.29643 (C_34_H_48_O_17_, −4.986 ppm). The ESI−MS/MS spectra had fragment ions at *m*/*z* 127.03926, *m*/*z* 189.08713, *m*/*z* 156.07675, *m*/*z* 213.09970, and *m*/*z* 159.07664. In the negative ion mode, M17 eluted at 15.70 min and had an [M − H]^−^ ion at *m*/*z* 727.28078 (C_34_H_48_O_17_, −0.135 ppm). Fragmentation resulted in C–O bond cleavage and the formation of ions at *m*/*z* 217.01762, *m*/*z* 261.00748, *m*/*z* 213.00554, and *m*/*z* 137.06081. Therefore, M17 was identified as a reduction product of dihydroxy–DISS.

The results of metabolomics analysis of DISS, allowed for the construction of a metabolic pathway diagram (Figure 7). Red compounds indicated greater distributions in the biological samples of the sham control group compared with those in the model group. Metabolites M0, M1, and M14 were detected in the feces sample, M9 was present in the brain, and M13 was present in the kidney of the sham control group. In contrast, M0, M1, and M14 were not detected in the feces sample of the model group, M9 was not detected in the brain, and M13 was not detected in the kidney. The green compounds were detected in the biological samples obtained from the model group. The metabolites M4 and M19 passed through the blood–brain barrier of the model group, but not the sham control group. Furthermore, M6 was detected in the urine, kidney, and liver of the model group, but not the sham control group. Metabolite M7 was detected in the liver and kidney, and M9 was detected in the urine and feces of the model group. Differential production of metabolites can highlight the mechanisms by which DISS exerts neuroprotective effects, and future studies can focus on the determination of structural correlates of the neuroprotective effects of M4 and M19.

## 4. Discussion

The NORT is a behavioral test that uses mouse exploratory behavior to evaluate learning and spatial memory [30]. The NORT allows the animal to freely explore the space without external stimuli, and the results are summarized using the Novel Object Recognition Index. The MWM experiment, a classic test for the evaluation of the learning and spatial memory abilities of AD model animals, utilizes the natural swimming ability of mice to find the location of a hidden platform in the water following training to the location of the platform [31,32]. In our study, the NORT and MWM were used to determine whether the model group of mice overexpressing APPswe_695_ had impaired learning and spatial memory capabilities. The tendency to explore a novel object was significantly reduced in model mice compared with that in the sham control group. Furthermore, learning, memory, and spatial memory were significantly impaired in the model group compared with those in the sham control group. These results were consistent with symptoms of AD, which was further confirmed by Nissl staining. There were significantly less hippocampal neurons in the model group. Based on confirmation of the model, differences in the metabolism of DISS between the groups were evaluated.

Liang et al. [33] detected and identified 10 metabolites of DISS in rats, including 3′-myrosinoyl sucrose, 6′-myrosinoyl sucrose, sinapic acid, and a series of ester bond cleavage products. Based on these findings, we characterized the metabolites of DISS in AD model mice for the first time, and successfully summarized the differential metabolic profiles of DISS in AD vs. sham control animals. The main metabolites of DISS in vivo, mainly occurred as ester bond cleavage, glycosidic bond cleavage, demethylation, and demethoxylation. After gavage of DISS in mice, the glycosidic bond and C–O bond were gradually broken and decomposed several times, forming M1, M2, M3, M8, M15 and M16. Under the action of methyltransferase, the demethylation formed the metabolites M4, M5, M9, M12, M14 and M18. The metabolites were further decomposed with dehydroxyl and some complex reactions, forming M6, M7, M10, M11, M13, M17 and M19. Finally metabolites entered the intestine through the hepatoenteric circulation and were excluded from the body. Most metabolites were found in the plasma, urine, and feces, and nearly half of the metabolites were undetected in the liver and kidney. Some small molecules, such as M1, M2, M3, M4 and M19 can enter the blood, and M4 and M19 can pass through the blood–brain barrier of model mice, which can be important active components of DISS.

The results showed that M0 (DISS), M1 (6′-sinapoyl sucrose), and M14 (DISS glycosidic bond rupture product) were detected in the feces of the sham control group, but were not present in the model group, which indicated that DISS had broken glycosidic bonds in vivo and produced M1 and M14, and DISS, M1 and M14 were better absorbed and distributed in the model group. Furthermore, the metabolites M4 and M19 were the demethoxide products of DISS, and transported across the blood–brain barrier to the brain, which indicates that these can be active metabolites responsible for the neuroprotective effects of DISS. In addition, the metabolites M6 and M9 were found to be differentially distributed in the model and sham control groups of mice. In summary, M0, M1, M4, M6, M9, M14 and M19 were differentially distributed in the model compared to the sham control group, and can be responsible for some of the pharmacological activities of DISS. The further study of these metabolites can improve our understanding of the biological functions of DISS.

## 5. Conclusions

In this study, Nissl staining, NORT, and MWM behavioral tests were used to determine the modeling status of AD mice, and UHPLC–Orbitrap mass spectrometry was used to detect and identify metabolites of DISS in mice. Twenty metabolites of DISS were either tentatively or positively identified, through using high mass accuracy precursor and fragmentation data. The biotransformation pathways of DISS in mice included ester bond cleavage, demethoxylation, and demethylation, and the metabolites M4 and M19 were shown to cross the blood–brain barrier in AD mice, which can indicate the neuroprotective mechanisms of DISS. As DISS exerts various pharmacological activities, such as neuronal protection and the alleviation of AD symptoms, our study provides a valuable reference for the further investigation of the mechanisms of action and biosafety of DISS.

## Figures and Tables

**Figure 1 molecules-27-00114-f001:**
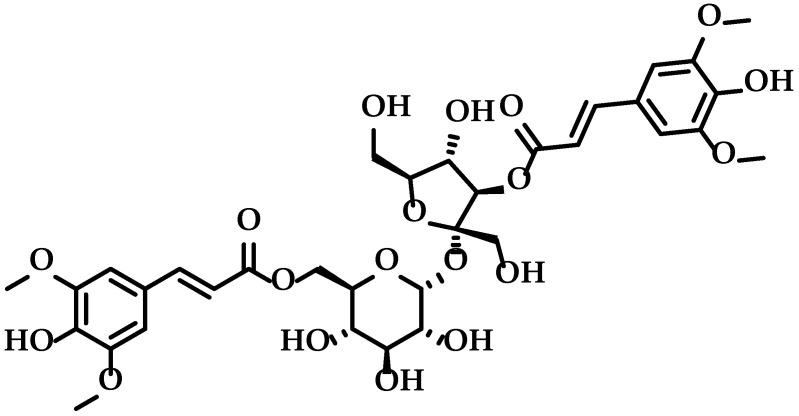
Structure of 3,6′-disinapoyl sucrose.

**Figure 2 molecules-27-00114-f002:**
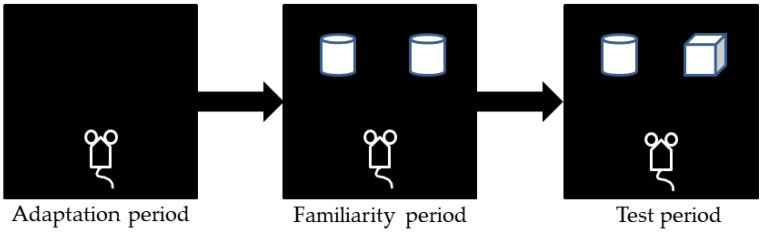
Novel object recognition test experimental procedure in sham control group (*n* = 10) and model group (*n* = 12).

**Figure 3 molecules-27-00114-f003:**
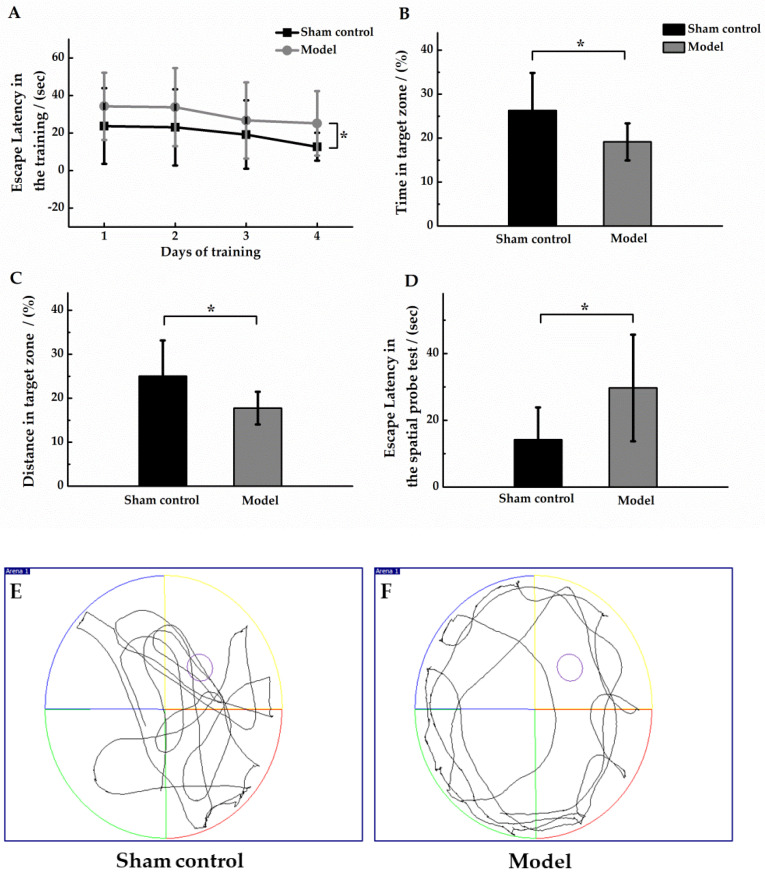
Morris Water Maze test in sham control group (*n* = 10) and model group (*n* = 12). (**A**) Mean escape latency for both groups of animals. Training was performed three times per day for four days. In the spatial probe test, the ratio (%) of the residence time in the target area to the total time (**B**); the ratio (%) of the distance swam by the animal in the target area to the total distance swam (**C**); and the escape latency was defined as the time when the animal first crossed the location of the hidden platform (**D**). Compared with the sham control group, * *p* < 0.05. In the spatial probe test, the movement trajectories (60 s duration) of one mouse in the sham control group (**E**) and model group (**F**) were shown, and the platform was unavailable for escape.

**Figure 4 molecules-27-00114-f004:**
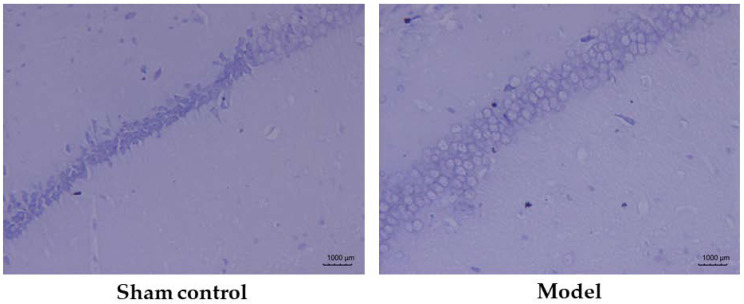
Pathological morphology of the hippocampal CA1 region in the sham control group (*n* = 10) and model group (*n* = 12).

**Figure 5 molecules-27-00114-f005:**
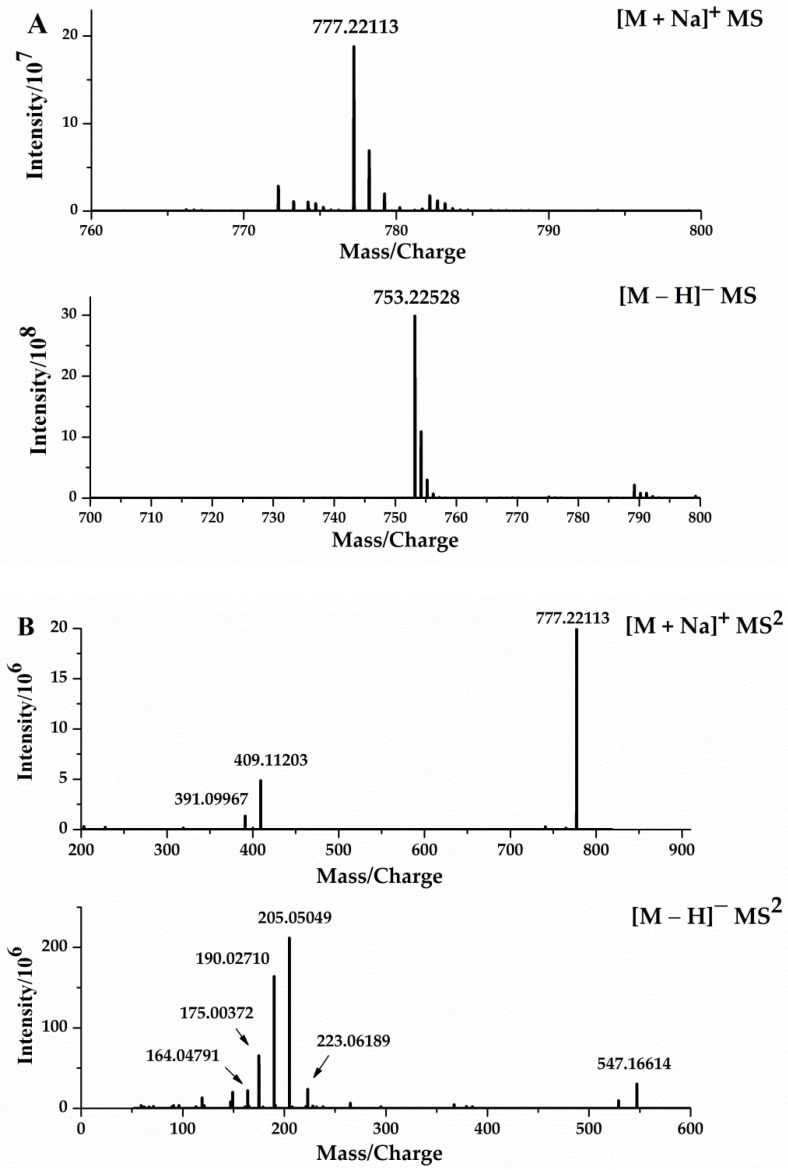
ESI–MS^1^ spectra (**A**), ESI–MS^2^ spectra (**B**) of DISS in positive and negative modes, the fragmentation behaviors of DISS in positive ion mode (**C**), and negative ion mode (**D**).

**Figure 6 molecules-27-00114-f006:**
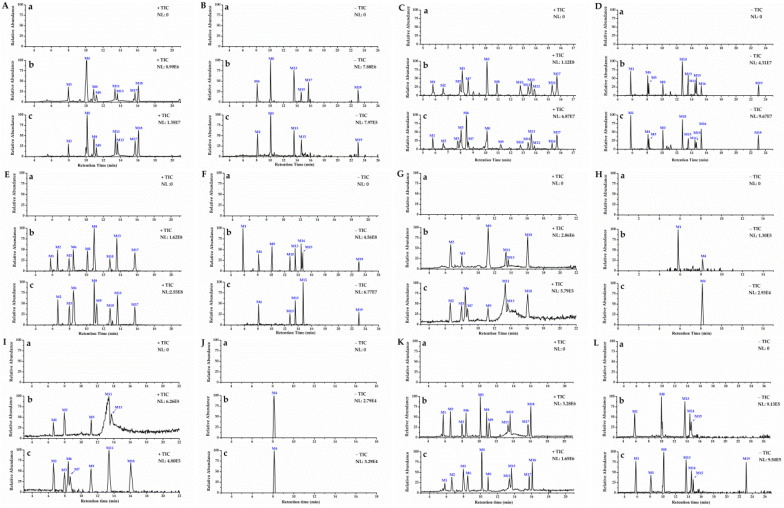
High-resolution TICs of blank group (a), and metabolites in biological samples of sham control group (b) and model group (c). TICs of plasma samples in positive mode (**A**) and negative mode (**B**); urine samples in positive (**C**) and negative mode (**D**); feces samples in positive (**E**) and negative mode (**F**); liver samples in positive (**G**) and negative mode (**H**); kidney samples in positive (**I**) and negative mode (**J**) and brain samples in positive (**K**) and negative mode (**L**).

**Figure 7 molecules-27-00114-f007:**
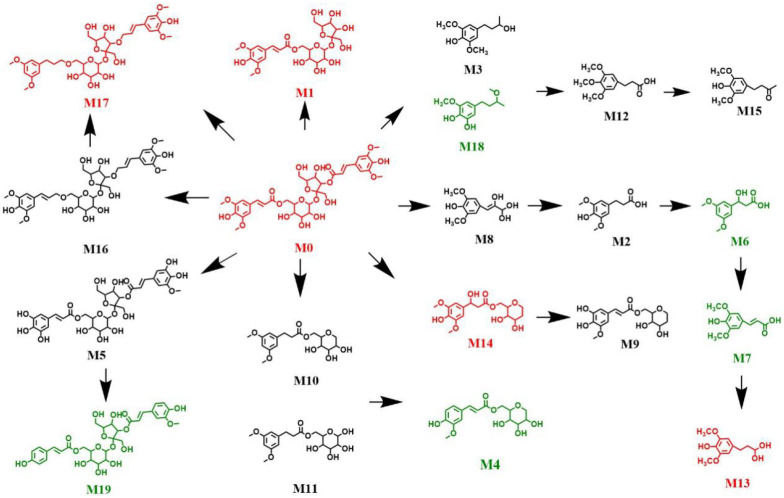
Metabolic pathways of DISS in mice.

**Table 1 molecules-27-00114-t001:** Summary of DISS metabolites by UHPLC–Orbitrap.

Peak	T_R_/min	Formula	Exact Mass	Error (ppm)	Adduct Ion (*m*/*z*)	MS^2^ Fragment (*m*/*z*)	Identification/Reactions	Distribution	
								Sham Control Group	Model Group
M0	10.12	C_34_H_42_O_19_	754.23148	−0.309	[M + Na]^+^ 777.22101	777.22113 (C_34_H_42_O_19_Na, 100), 409.11023 (C_17_H_22_O_10_Na, 21.58), 391.09967 (C_17_H_20_O_9_Na, 6.09)	DISS	1, 2, 3, 6	1, 2, 6
	10.11	C_34_H_42_O_19_	754.23148	2.157	[M − H]^−^ 753.22528	205.05049 (C_11_H_9_O_4_, 100), 190.02710 (C_10_H_6_O_4_, 88.42), 175.00372 (C_9_H_3_O_4_, 28.26), 547.16614 (C_23_H_31_O_15_, 12.27), 164.04791 (C_9_H_8_O_3_, 8.76), 223.06189 (C_11_H_11_O_5_, 7.23)		1, 2, 3, 6	1, 2, 6
M1	5.78	C_23_H_32_O_15_	562.15284	−0.773	[M + Na]^+^ 571.16290	571.16266 (C_23_H_32_O_15_Na, 100), 409.10941 (C_17_H_22_O_10_Na, 22.62), 391.10001 (C_17_H_20_O_9_Na, 6.90)	6′-Sinapoyl sucrose	2, 3, 6	2, 6
	5.78	C_23_H_32_O_15_	562.15284	1.724	[M − H]^−^ 547.16669	205.05058 (C_11_H_9_O_4_, 100), 190.02713 (C_10_H_6_O_4_, 57.13), 175.00354 (C_9_H_3_O_4_, 20.99), 547.16681 (C_23_H_31_O_15_, 18.35), 223.06117 (C_11_H_11_O_5_, 11.96)		2, 3, 4, 6	2, 6
M2	6.61	C_11_H_14_O_5_	226.08357	−0.529	[M + H]^+^ 227.09128	181.04924 (C_9_H_9_O_4_, 100), 227.09102 (C_11_H_15_O_5_, 26.19), 209.08060 (C_11_H_13_O_4_, 21.94), 167.07001 (C_9_H_11_O_3_, 6.11)	C–O bond rupture reduction products of DISS	2, 3, 4, 5, 6	2, 3, 4, 5, 6
M3	7.96	C_12_H_18_O_4_	226.11996	−0.553	[M + H]^+^ 227.12766	85.06531 (C_5_H_9_O, 100), 149.09598 (C_10_H_13_O, 63.70), 191.10643 (C_12_H_15_O_2_, 61.59), 209.11690 (C_12_H_17_O_3_, 60.39), 227.12654 (37.93)	Methylation product of M2	1, 2, 3, 4, 5, 6	1, 2, 3, 4, 5, 6
M4	8.13	C_16_H_20_O_8_	340.11527	3.792	[M − H]^−^ 339.10873	339.05457 (C_11_H_15_O_12_, 100), 259.09747 (C_15_H_15_O4, 57.77), 229.08693 (C_14_H_13_O_3_, 55.89), 122.03733 (C_7_H_6_O_2_, 9.81), 134.03729 (C_8_H_6_O_2_, 7.20), 295.09653 (C_18_H_15_O_4_, 5.07)	DISS demethoxylation product	1, 2, 3, 4, 5	1, 2, 3, 4, 5, 6
M5	8.14	C_31_H_36_O_19_	712.18453	0.1914	[M + Na]^+^ 735.17440	70.06594 (100), 245.11336 (C_13_H_18_O_3_Na, 41.27), 227.10280 (C_13_H_16_O_2_Na, 36.82), 277.11856 (C_15_H_15_O_2_, 32.87), 249.12337 (C_16_H_18_ONa, 27.44), 325.17694 (C_19_H_26_O_3_Na, 26.45), 343.18710 (C_19_H_28_O_4_Na, 24.09), 201.12357 (C_12_H_18_ONa, 21.03)	Demethylations product of DISS	2	2
	8.13	C_31_H_36_O_19_	712.18453	2.299	[M − H]^−^ 711.17834	321.04388 (C_11_H_13_O_11_, 100), 241.08717 (C_15_H_13_O_3_, 25.58), 322.04718 (C_18_H_10_O_6_, 13.30), 294.13840 (C_23_H_18_, 10.90)		2	2
M6	8.46	C_11_H_14_O_5_	226.08357	−0.485	[M + H]^+^ 227.09129	167.07014 (C_9_H_11_O_3_, 100), 177.05444 (C_10_H_9_O_3_, 5.15), 209.08076 (C_11_H_13_O_4_, 3.85)	C–O bond rupture reduction products of DISS	3, 6	2, 3, 4, 5, 6
M7	8.61	C_11_H_12_O_5_	224.06792	−0.222	[M + H]^+^ 225.07570	207.06502 (C_11_H_11_O_4_, 100), 175.03889 (C_10_H_7_O_3_, 63.01), 192.04141 (C_10_H_8_O_4_, 21.64), 149.02347 (C_8_H_5_O_3_, 7.54), 164.04666 (C_9_H_8_O_3_, 5.36)	Sinapinic acid	2	2,4,5
M8	10.90	C_11_H_14_O_6_	242.07849	1.921	[M + H]^+^ 243.08733	243.08736 (C_11_H_15_O_6_, 100), 172.08670 (40.40), 216.07651 (29.79), 198.06587 (15.69), 197.08096 (C_10_H_13_O_4_, 0.46)	Hydroxylation of Sinapic acid	1, 2, 3, 6	1, 2, 3, 6
M9	11.21	C_16_H_20_O_8_	340.11527	−4.438	[M + H]^+^ 341.12158	341.22040 (100), 238.19051 (C_13_H_27_O_2_Na, 4.07), 205.08562 (C_11_H_9_O_4_, 2.72), 323.09164 (C_19_H_15_O_5_, 2.71), 191.07014 (C_11_H_11_O_3_, 2.70), 107.08558 (C_8_H_11_, 2.32), 163.07523 (C_10_H_11_O_2_, 2.16)	DISS demethoxylation product	1, 4, 5, 6	1, 2, 3, 4, 5
M10	12.77	C_17_H_24_O_9_	372.14148	−1.856	[M + Na]^+^ 395.13052	395.28958 (C_25_H_40_O_2_Na, 100), 377.27933 (C_25_H_38_ONa, 36.24), 107.08579 (C_6_H_12_Na, 9.68), 271.12012 (C_11_H_20_O_6_Na, 5.33), 359.26834 (C_25_H_36_Na, 5.17)	DISS glycosidic bond rupture product	2, 3	2, 3
	12.75	C_17_H_24_O_9_	372.14148	3.399	[M − H]^−^ 371.13492	147.11789 (100), 371.17041 (C_18_H_27_O_8_, 67.35), 209.11717 (C_12_H_17_O_3_, 4.69), 179.05547 (2.83), 191.10738 (2.68), 149.04442 (C_5_H_9_O_5_, 2.16), 293.94162 (1.78), 207.10025 (1.73)		2, 3	2, 3
M11	13.39	C_17_H_24_O_9_	372.14148	−0.388	[M + Na]^+^ 395.13120	395.20325 (C_19_H_32_O_7_Na, 100), 203.05223 (C_6_H_12_O_6_Na, 23.02), 201.03685 (C_6_H_10_O_6_Na, 6.26), 217.15601 (C_13_H_22_ONa, 5.52), 215.14053 (C_13_H_20_ONa, 4.39), 121.10120 (C_9_H_13_, 1.80)	DISS C=O bond fracture product	1, 2, 4, 5, 6	1, 2, 4, 5, 6
M12	13.90	C_12_H_16_O_5_	240.09922	0.456	[M + H]^+^ 241.10716	241.16249 (100), 123.116687 (C_9_H_15_, 25.31), 147.11676 (C_11_H_15_, 13.58)	Methylated product after reduction of sinapinic acid	2	2
M13	13.56	C_11_H_16_O_5_	228.09923	−0.338	[M + Na]^+^ 251.08891	98.98458 (100), 251.08862 (C_11_H_16_O_5_Na, 76.53), 251.12540 (C_12_H_20_O_4_Na, 61.89), 141.09071 (C_6_H_14_O_2_Na, 7.51), 167.08630 (C_11_H_12_Na, 6.81), 173.13144 (C_11_H_18_Na, 5.06)	Sinapinic acid addition product	1, 2, 3, 4, 5, 6	1, 2, 3, 4, 6
	13.52	C_11_H_16_O_5_	228.09923	4.976	[M − H]^−^ 227.09253	227.12891 (C_12_H_19_O_4_, 100), 183.13898 (C_11_H_19_O_2_, 30.81), 165.12851 (C_11_H_17_O, 27.80), 209.11829 (C_12_H_17_O_3_, 25.19)		1, 2, 3, 6	1, 2, 3, 6
M14	14.48	C_17_H_24_O_9_	372.14148	−3.418	[M − H]^−^ 371.13239	371.11700 (C_13_H_23_O_12_, 100), 291.16016 (C_17_H_23_O_4_, 66.46), 79.95737 (30.93), 247.17033 (C_16_H_23_O_2_, 30.14), 371.24493 (24.66), 122.03730 (17.26), 123.04517 (13.98), 135.04510 (13.06)	DISS glycosidic bond rupture product	2, 3, 6	2, 6
M15	14.64	C_12_H_16_O_4_	224.10431	3.382	[M − H]^−^ 223.09724	223.11705 (C_9_H_19_O_6_, 100), 122.10566 (C_9_H_14_, 13.98), 179.12711 (C_8_H_19_O, 44.85)	Reductive product of M12	1, 2, 3, 6	1, 2, 3, 6
M16	15.32	C_34_H_46_O_17_	726.27295	−1.947	[M + H]^+^ 727.28077	409.09137 (C_22_H_17_O_8_, 100), 427.10205 (C_22_H_19_O_9_, 98.04), 391.08096 (C_22_H_15_O_7_, 72.55), 379.08099 (C_21_H_15_O_7_, 71.11), 511.12332 (C_26_H_23_O_11_, 50.98), 481.11246 (C_25_H_21_O_10_, 50.12), 325.07050 (C_18_H_13_O_6_, 40.00), 349.07037 (C_20_H_13_O_6_, 33.57), 337.07034 (C_19_H_13_O_6_, 24.35), 355.08121 (C_19_H_15_O_7_, 23.50)	DISS reduction product	2	2
	15.30	C_34_H_46_O_17_	726.27295	1.2745	[M − H]^−^ 725.26600	353.06680 (100) 443.09808 (C_22_H_19_O_10_, 76.65), 473.10849 (C_23_H_21_O_11_, 34.99), 725.19037 (C_32_H_37_O_19_, 33.48), 545.12988 (C_26_H_25_O_13_, 18.62), 413.08752 (C_21_H_17_O_9_, 15.06), 407.07764 (C_22_H_15_O_8_, 11.06)		2	2
M17	15.69	C_34_H_48_O_17_	728.28860	−4.986	[M + H]^+^ 729.29643	127.03926 (C_6_H_7_O_3_, 100), 189.08713 (C_12_H_13_O_2_, 53.89), 156.07675 (C_8_H_12_O_3_, 41.81), 155.08171 (C_12_H_11_, 19.55), 213.09970 (C_7_H_17_O_7_, 16.21), 159.07664 (C_11_H_11_O, 15.36)	Reduction products of dehydroxy DISS	1, 2, 3, 6	1, 2, 3, 6
	15.70	C_34_H_48_O_17_	728.28860	−0.135	[M − H]^−^ 727.28078	212.00232 (100), 217.01762 (C_4_H_9_O_10_, 43.23), 80.96514 (21.35), 261.00748 (C_12_H_5_O_7_, 17.73), 213.00554 (C_8_H_5_O_7_, 10.55), 137.06081 (C_8_H_9_O_2_, 10.27)		1	−
M18	16.04	C_12_H_18_O_4_	226.11996	0.5040	[M + H]^+^ 227.1279	227.06352 (C_12_H_12_O_3_Na, 100), 209.11656 (C_12_H_17_O_3_, 71.14)	Position isomer of M3	1, 4, 6	1, 4, 5, 6
M19	23.05	C_31_H_36_O_16_	664.18453	−1.938	[M − H]^−^ 663.17542	112.98559 (100), 323.22034 (C_19_H_31_O_4_, 32.70), 255.23296 (C_16_H_31_O_2_, 25.16), 391.20743 (C_29_H_27_O, 10.03)	Demethylations product of DISS	1, 2, 3	1, 2, 3, 6

Note: t_R_: retention time; 1: plasma; 2: urine; 3: feces; 4: liver; 5: kidney; 6: brain; and +: detected.

## Data Availability

Data that support the findings of this study are available from the corresponding author upon reasonable request.

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
