# Peer review of "Rapid Identification of 3,6′-Disinapoyl Sucrose Metabolites in Alzheimer’s Disease Model Mice Using UHPLC–Orbitrap Mass Spectrometry"

_molecules, 2021, doi:10.3390/molecules27010114_

Round 1
Reviewer 1 Report
Please revise and correct for errors in written English.
Author Response
We revised the manuscript, thank you.
Reviewer 2 Report
This manuscript presents the elucidation of the structures of metabolites of 3,6’-disinapoyl sucrose (DISS), a key component of the traditional Chinese medicine Kaixin San known for its beneficial impact on dementia. The research was conducted in ICR mice with weights of 19-21 g, and the metabolites were identified in the blood, urine, feces, brain, kidneys and liver of the model group animals using reversed-phase ultra-high performance liquid chromatography hyphenated with high resolution mass spectrometry following their preparation using solid phase extraction (SPE) and solvent exchange which included the concentration of samples. The animal model consisted of the intra-hippocampal injection of a plasmid containing the APPswe695 gene (model group) or a blank plasmid (sham control group), or no plasmid (blank control group). The animals underwent a novel object recognition test (NORT) and a Morris Water Maze (MWM) test, followed by blood sampling from the suborbital venous plexus on 6 occasions 0.5-6 h after the oral administration of 200 mg/kg DISS in a solution. The authors found that the range of metabolites detected in the pooled samples of the group of animals with a triggered Alzheimer’s disease was considerably vaster than that observed in the sham group (receiving a plasmid solution which did not contain the gene triggering Alzheimer’s). The results of behavioral tests are presented in detail as well as those regarding the identification, location and elucidation of the metabolites detected. The contents of the manuscript are supported by 33 references.
I understand that the design of this work was based on a previous study published in the same journal (Reference [2]). Still, the topic of the manuscript is of limited interest since the results presented in it are preliminary – no evidence of the biological effects of any of the elucidated compounds, or their relation to the results of the NORT and MWM tests is shown. The main value of this work is that a very sophisticated analytical system was employed which lends much credit to the proposed chemical structures and to the presented metabolic pathway of DISS. Nevertheless, there are several issues which require clarification and correction.
Major remarks:
- The following serious doubt regarding the credibility of the research is raised by the described methodology of the animal experiments. 1 mL plasma was applied to the SPE cartridges (Section 2.6.4, line 178), which was a pool of samples collected from 12 animals. The mice who were the source of these samples were of weights 19-21 g (Section 2.2, line 94), with a total blood volume of 1.1-1.4 mL each (see for instance the Guidelines for Blood Collection in Mice and Rats, National Institutes of Health, United States, https://oacu.oir.nih.gov/system/files/media/file/2021-02/b2_blood_collection_in_mice_and_rats.pdf, accessed 12 November 2021). In section 2.6.1, Lines 151-152, the authors claim that sampling was performed at 0.5, 1, 1.5, 2, 4 and 6 h postdose, 6 occasions altogether. Therefore, it cannot have been possible to collect the required volume of blood (more than 9 mL whole blood, i.e. 0.75 mL blood from each animal corresponding to 54-68% of their total circulating blood volume) in such a short amount of time without killing, but at the very least without very seriuosly altering the physiological status of the animals. In short, it is inconceivable that the experiments were performed in the way they are described. The authors should address this in full detail.
- In addition to the above, although the authors claim there were 6 occasions of sampling per animal, they fail to even mention why this was important. If the required volume was 1 mL for analysis, why were so many animals used, and so many sampling times needed? Why did not the authors use e.g. 3 animals per experimental group, sacrifice them, and remove a large volume of blood from each along with the dissection and processing of organs at a single time point? Please provide the results obtained at the 6 sampling times and discuss them.
- Since the samples obtained in a specific experimental group of animals were combined, the number of the animals in a group which the presented metabolites could be detected in is not clear. The authors should provide evidence that the range of the detected metabolites were observed in more than one or some of the animals belonging to the Alzheimer group to establish the biological significance of the findings.
- The methodology described for the NORT and MWM experiments lacks detail which would lend credibility, comparability and reproducibility of the observations, and so do the results presented. Table 1 contains very few data which should be included in the text instead. The data shown in Figure 2 should be supported by detailed numerical findings in the specific observations, at least provided as supplementary information. It is important to describe how the obervations were made: were they subjective findings made by the laboratory personnel, or did the authors employ an apparatus? If so, what was it?
- The language of the manuscript must clearly be improved as there are numerous sentences which cannot be understood clearly. Some examples:
- „The optimized MS analysis operating parameters was carried out by the ESI source” (Section 2.8, lines 206-207) - e,
- „Feces samples were dried and grounded, and 1 g feces sample of 2 mL physiological saline were homogenized and ultrasonic extraction with ethyl acetate (1 mL) for three times, then collected ethyl acetate.” (Section 2.6.2, lines 158-161)
- „The supernatants of each group were gathered as the blank sample and test samples. With this mothed (sic), the liver and kidney samples were prepared.” (Section 2.6.3, lines 172-174)
- No description of the employed statistical methods is provided, regardless of the title of Section 2.9.
- In Figure 5, the total ion chromatograms of the metabolites in the collected urine of the model animals and the sham animals are shown. The total ion chromatogram obtained in the group of animals providing the ’blank’ samples is lacking. Chromatograms obtained in plasma, feces, kidney and liver are also lacking. These would be extremely important for facilitating the understanding of the presented work.
- Compared to the complexity of the manuscript (i.e. results are presented regarding behavioral experiments as well as bioanalysis including the chemical structure elucidation of several substances), the Discussion section is very brief. Please discuss the findings in more detail.
Minor remarks:
- In Section 2.2 (line 94), the authors say 32 animals were used. All three experimental groups included 12 animals (line 98), which totals up to 36 animals. Please rectify this discrepancy.
- SPE is an extraction method. The authors are wrong when stating it is a method employed to precipitate proteins and solid residues (Section 2.6.4, lines 176-177).
- The authors claim that the feces ethyl acetate extracts and urine samples were centrifuged to absorb supernatants (Section 2.6.2, lines 161-162). Centrifugation was probably performed to separate supernatants from the solid content of these samples, not to absorb them.
- The authors claim the SPE cartridges were pre-activated with 5 mL methanol and 5 mL water (Section 2.6.4, lines 177-178). After applying the plasma samples, the cartridges were eluted using 5 mL water (lines 178-179). This requires clarification since the volume of the barrels was 3 mL (Section 2.1, line 89).
- Lines 196-198: „The UHPLC separation was carried out […] equipped with a diode array detector (DAD) to identify metabolites.” The metabolites were identified using Orbitrap MS, not DAD. Please be specific on the use of DAD if it had any importance.
- Sample availability has not been specified (Line 461).
Reviewer 3 Report
This is a very good publication. It deserves to be published in Molecules, although it could just as easily have been published in Metabolites (and I think that it would have been a much better journal for the authors, but I understand and respect their decision to publish in Molecules). The subject matter is very topical and all the experiments were carried out very correctly. I only have a few minor questions:
Why were fecal and urine samples combined in each group?
On what basis/how were the sample preparation methods of each matrix chosen? Were the extraction conditions chosen in any way (e.g. why ethyl acetate was used for ultrasonic extraction; why was the C18 cartridge chosen for SPE)?
Were the MS operating parameters selected/optimized?
These comments do not diminish the value of the work, which I rate highly.
Round 2
Reviewer 2 Report
Figures:
Figure 2 caption is missing. The meaning of the asterisks is not clear. Please insert the number of animals (n) used in the sham control and the model groups.
Figure 3: ** P<0.01 is given in the caption, but no differences indicated with ** appear in the Figure. Figure 3 E and F: Are these trajectories showing the movement of one or more animals? Please use recognizably different lines of each animal as in this form the plot is undecipherable. Please insert the number of animals (n) used in the sham control and the model groups.
Figure 6: Please provide the chromatograms of all of the matrices explored (plasma, feces, brain, kidney and liver) in the Figure. There is apparently no reason to pick urine as a matrix of outstanding importance. There are two (A) and (B) chromatograms in the Figure, please correct this.
Tables:
Table 2: Caption - UPLC should be replaced with UHPLC as this abbreviation is used throghout the manuscript. Lien 390: I recommend moving the contents of the Note to the caption for a better understanding.
Manuscript:
Lines 117-119: In the revision, the authors have modified the number of animals used from 36 to 32. In addition, they have provided the sizes of the groups. Please specify the rationale behind dividing the animals in the various groups as described. The identified metabolites come from an exogenously administered substance. What endogenous interference did the authors expect which prompted them to use 10 animals for this purpose? What was the reason for using a different number of animals in the sham control and the model groups? Looking at these numbers, one does not see a reason for not using the same sample sizes at least for the groups participating in the behavioral experiments.
Line 189: The abbreviation ’DISS’ has been explained already in line 74.
Lines 195-197: I don’t see a rational relationship between the absorption maximum of DISS at 90 min, its half-life of 5 h, and the 0-6 h sampling period. I think in fact the sampling period fell between 0.5-6 h as the first sample was drawn 0.5 h postdose. A half-life of 5 h would make sampling at later time points, at least at 10 or 15 h (2 and 3 half-lives, respectively) more reasonable. I also don’t see how this would be related to the formation of the metabolites. Is there a reason against the assumption that some metabolites, not detected by the authors, may have appeared at a later time point (especially in view of the fact that the samples collected at different time points were pooled, therefore, the kinetic properties of the metabolites remained unelucidated)? Please address this dilemma.
Line 195: I recommend using the expression ’time to reach peak concentration’ instead of ’absorption maximum’.
Line 198: Please specify that the plasma samples were pooled separately for the sham control and for the model group.
Lines 201-203: The blood sampling protocol is still obscure. Please provide the details, including which animal donated blood at which time point, as well as the volume of the samples collected.
Sections 2.6.4 and 2.6.5 (Lines 223-246): The authors say that the feces and tissue homogenates were extracted using ethyl acetate. The organic extracts were subjected to the same sample preparation procedure (in my understanding, solid phase extraction using octadecylsilica packing) as plasma and urine. Ethyl acetate, being a hydrophobic solvent, is not used for sample loading onto C18 packings. The procedure described in lines 237-241 is clearly one suitable for extracting substances from aqueous media (pre-conditioning and washing was performed using water). Please revise the sample preparation employed for recovering the analytes from feces, liver, brain and kidney as the procedure described in the manuscript is certainly not one that can be successfully performed.
Lines 229-231: The authors say approximately 0.1 g tissue was pooled. How approximately? Please provide the ranges of the masses of the tissues used. In addition, please be explicit that pooling of the samples of the animals belonging to the sham control and the model groups was performed separately (if they in fact were).
Line 240: Elution with 5 mL water is not feasible. Was it 2x2.5 mL?
Line 283, 288: Please provide the sources of these softwares (SPSS, Origin).
Lines 286-287: The authors say that Bonferroni test was performed to determine significant differences between groups. If this means that Student t-tests were performed with the Bonferroni correction, please put it this way for easier understanding. Also, in line 286, the general term ’other experimental data’ should be replaced with the specific types of the experimental results this test was applied to.
Line 289: Please specify that the data collected were mass spectrometry data.
Line 307: Table 1 contains very little information. I recommend deleting the table and moving its contents to the text.
Line 313: P<0.01 is written, but in Figure 2A, a single asterisk (*) is used for indicating the level of the difference. Please be consistent on indicating the levels of significance (e.g. for P<0.05 and ** for P<0.01) in the Figures.
Lines 533 and 534: „In contrast, these metabolites were not detected in the model group.” – The data provided in Table 2 contrast this statement. M0 was found in plasma, urine and feces, M1 in urine and brain, M9 in all samples except brain, and so on, in the model group. Please correct this sentence.
Lines 536-538: Metabolites M4 and M19 passed through the blood-brain barrier of the model group animals, but not of the sham control group animals. What is the rationale for this? Since samples were pooled, it would be important to emphasize that the presence of any of the substances in as few as 1 of the animals in a group has been identified as a presence in the sham control or model group, and, therefore, that inferring any conclusions of biological relevance requires more detailed research.
Lines 574-590: The authors explored the presentation of DISS and its metabolites in blood, urine, feces, brain, kidney and liver. The discussion of their findings in the various organs is very brief. Please give a detailed discussion of the associations among the metabolites identified in the various types of samples.
